Spatio-temporal causal graph attention network for traffic flow prediction in intelligent transportation systems

Zhao Wei 1 2 3
Zhang Shiqi 3 zsq_alan@163.com
Wang Bei 3
Zhou Bing 1 2
1 School of Artificial Intelligence and Computer Science, Zhengzhou University , Zhengzhou , China
2 Cooperative Innovation Center of Internet Healthcare, Zhengzhou University , Zhengzhou , China
3 School of Cyber Science and Engineering, Zhengzhou University , Zhengzhou , China
Stević Željko
Electronic publication date: 2023 Jul 28
Publication date: 2023
Volume: 9
Electronic Location ID: e1484
Received 2023 Apr 17; Accepted 2023 Jun 16
Copyright: © 2023 Zhao et al.
Copyright year: 2023
Copyright holder: Zhao et al.
License: This is an open access article distributed under the terms of the Creative Commons Attribution License, which permits unrestricted use, distribution, reproduction and adaptation in any medium and for any purpose provided that it is properly attributed. For attribution, the original author(s), title, publication source (PeerJ Computer Science) and either DOI or URL of the article must be cited.
License URL: https://creativecommons.org/licenses/by/4.0/

Keywords: Traffic flow prediction, Intelligent transportation systems, Artificial intelligence, Graph convolution neural networks, Time series prediction

Funding: National Key Technologies R&D Program 2020YFB1712401, 2018YFB1701400 Henan Province of China 201300210500 Nature Science Foundation of China 62006210 Nature Science Foundation of China 62006210 Research Foundation for Advanced Talents of Zhengzhou University 32340306 Key Project of Collaborative Innovation in Nanyang 22XTCX12001 This work was supported by the National Key Technologies R&D Program (2020YFB1712401, 2018YFB1701400), the 2020 Key Project of Public Benefit in Henan Province of China (201300210500), the Nature Science Foundation of China (62006210), the Nature Science Foundation of China (62006210), the Research Foundation for Advanced Talents of Zhengzhou University (32340306), and the Key Project of Collaborative Innovation in Nanyang (22XTCX12001). The funders had no role in study design, data collection and analysis, decision to publish, or preparation of the manuscript.

==============================
Accurately predicting traffic flow on roads is crucial to address urban traffic congestion and save on travel time. However, this is a challenging task due to the strong spatial and temporal correlations of traffic data. Existing traffic flow prediction methods based on graph neural networks and recurrent neural networks often overlook the dynamic spatiotemporal dependencies between road nodes and excessively focus on the local spatiotemporal dependencies of traffic flow, thereby failing to effectively model global spatiotemporal dependencies. To overcome these challenges, this article proposes a new Spatio-temporal Causal Graph Attention Network (STCGAT). STCGAT utilizes a node embedding technique that enables the generation of spatial adjacency subgraphs on a per-time-step basis, without requiring any prior geographic information. This obviates the necessity for intricate modeling of constantly changing graph topologies. Additionally, STCGAT introduces a proficient causal temporal correlation module that encompasses node-adaptive learning, graph convolution, as well as local and global causal temporal convolution modules. This module effectively captures both local and global Spatio-temporal dependencies. The proposed STCGAT model is extensively evaluated on traffic datasets. The results show that it outperforms all baseline models consistently.

Introduction

With the rapid development of the Industrial Internet of Things (IIoT) 4.0 (Majid et al., 2022), 5G, and even future 6G-based high-traffic communications (Zhao, Askari & Chen, 2021), and Artificial Intelligence (AI), the range of intelligent city sub-sectors is expanding to serve service providers and citizens fully. Among these services, Intelligent Transportation Systems (ITS) (Guevara & Auat Cheein, 2020) play a critical role in providing outstanding services. ITS can provide real-time and accurate road traffic status information, location navigation services, personalized travel route planning, and other services. Traffic prediction plays a crucial role in intelligent transportation systems by utilizing traffic data to uncover spatiotemporal patterns and identify potential traffic patterns. It can effectively mine potential Spatio-temporal patterns from traffic data. This not only helps to relieve traffic congestion and control traffic flow scheduling but also reduces people’s travel time and cost while reducing environmental pollution to promote the development of smart cities (Richter et al., 2020).

In order to make precise predictions about traffic patterns, it is essential to take into account both the temporal relationships between historical data points and the spatial relationships between nodes on the road network. In the past, traffic flow prediction methods mainly focused on analyzing multivariate series data in the time dimension. These approaches included modeling traffic time series data using queuing theory models (Xu et al., 2014), traffic behavior theory (Cascetta, 2013), and machine learning methods (Li & Xu, 2021). However, these methodologies only evaluated the correlation on the temporal dimension and disregarded the spatial dimension dependence. As a result, an increasing number of academics are focusing on Spatio-temporal prediction models based on graph neural networks (GNN) (Wu et al., 2020) that have shown impressive outcomes. However, these models still have some limitations that need to be addressed.

The first limitation of current Spatio-temporal prediction models based on GNN is that they ignore the dynamic correlation information between nodes on the graph. Typically, GNN-based spatial dependency modeling approaches use edge transformation and aggregation through the information of nodes in the traffic network (Wu et al., 2020). However, most of these (Li et al., 2017; Guo et al., 2019; Bai et al., 2020) approaches employ a predetermined static adjacency matrix to characterize the spatial interactions between traffic road nodes (Waikhom & Patgiri, 2021). This method overlooks the fact that the relationships between road nodes in a traffic network are dynamic and interact with each other, depending on various complex factors on the road, such as traffic flow, number of lanes, and population density. Therefore, modeling the spatial relationship of roads solely based on the static spatial connections between them is insufficient.

The second limitation of spatio-temporal traffic prediction models is that traffic information on the transportation network has a high degree of nonlinear correlation and uncertainty. For example, regular road maintenance and sudden accidents can significantly impact traffic flow. While recurrent neural networks (RNN) (Sha et al., 2020) such as long-short term memory (LSTM) (Karimzadeh et al., 2021) or Gated Recurrent Unit (GRU) (Sun, Boukerche & Tao, 2020) can capture temporal dependence, they have limitations when dealing with long-range sequences. The signals must traverse a long recurrent path of the network, making it challenging to model the global time dependence of long-time sequence data effectively. Additionally, the sequential execution process of RNNs makes it difficult for them to capture causal correlation information about traffic events (Li et al., 2020). To address these issues, some research approaches based on convolutional neural networks (CNN) stack convolutional layers into multiple layers to model global temporal dependencies (Zheng et al., 2020; Ma, Song & Li, 2020). However, as the expansion rate increases (Li & Zhu, 2021), local time-dependent information may be lost. Moreover, as the neural network’s depth increases, it becomes more challenging to optimize the model, leading to the problem of network degradation (He et al., 2016).

The proposed solution to address the aforementioned challenges is a novel Spatio-temporal prediction framework based on graph attention network (GAT) (Veličković et al., 2017) known as spatio-temporal causal graph attention network (STCGAT). The STCGAT framework offers a data-driven graph structure learning approach that autonomously learns the relationship information between road nodes and models the spatial correlation of traffic networks without relying on the traffic network graph structure information. Additionally, the article introduces a bidirectional spatio-temporal component to capture local and global spatial-temporal dependency information simultaneously. To capture more abstract spatio-temporal dependence information, a residual module is utilized within the component (He et al., 2016). The primary contributions of our work are: 1. A data-driven Node Adaptive Learning Graph Attention Network (NAL-GAT) is constructed in this study, which models the spatial dependence of traffic networks without the need for a predefined adjacency matrix.

2. A new spatiotemporal component is proposed in this study, wherein the GRU gating unit is replaced with NAL-GAT. This component is further designed as a recursive bidirectional network, aiming to capture the local causal spatiotemporal dependence.

3. The output sequence data is parallelly processed in this study by using a stacked temporal convolutional network, which aims to capture global and long-range temporal dependencies.

The structure of the document is as follows: In Section 2, a brief overview of relevant works in the field of spatio-temporal prediction is provided. Section 3 elaborates on the structure and methodology of STCGAT. In Section 4, the performance of STCGAT is evaluated by comparing it to various experimental outcomes. Finally, in Section 5, the work is summarized.

Literature review

Graph convolutional network

Graph convolutional networks (GCN) (Welling & Kipf, 2016) have gained significant attention in recent years due to their ability to process graph-structured data for various tasks, such as graph classification, node classification, and link prediction. GCN can be classified into two main categories based on their clustering method: Spectral-based and Spatial-based (Wu et al., 2020). Spectral-based techniques in graph signal processing use filters to define convolutions on graphs, and they extend these convolutions to the spectral domain by identifying the corresponding Fourier bases. This allows for efficient computation of convolutions on graphs and enables the use of tools from spectral graph theory to analyze graph signals. Some primary examples of spectral-based methods include Chebyshev Spectral CNN (ChebNet) (Monti et al., 2017) and adaptive graph convolution network (AGCN) (Li et al., 2018b). Spatial-based approaches define graph convolutions by aggregating feature information from the neighborhood and using the spatial relationship of the nodes. Some of the most prominent spatial-based methods include GAT and gated attention network (GAAN) (Zhang et al., 2018).

Time-dependent modeling

In the past, traffic forecasting projects relied on multivariate time series analysis methods such as history average (HA) (Duan et al., 2016), support vector regressor (SVR) (Hao, Leixiao & Hui, 2020), and vector autoregression (VAR) (Dissanayake et al., 2021). However, these techniques assume ideal smoothness and do not account for the nonlinear correlation in traffic data. In recent years, deep learning has gained popularity due to its ability to create sophisticated models and learn autonomously. Recurrent neural network (RNN)-based models have been widely used to capture temporal dependencies, but their long-loop structure is time-consuming and can lead to the gradient disappearance and explosion phenomenon. To overcome this issue, temporal convolutional network (TCN) (Bai, Kolter & Koltun, 2018; Zhang, Liu & Zheng, 2019) has been introduced to process time series in parallel and capture more information through a larger field of perception. Additionally, researchers have developed variant models based on Transform (Vaswani et al., 2017) that perform well on long time series prediction tasks (Xu et al., 2020; Zhou et al., 2021; Wu et al., 2021). These approaches solely take into account the temporal patterns of traffic flow data, disregarding the spatial interdependencies among the road nodes.

Spatio-temporal dependence modeling

Several studies have utilized regular two-dimensional grids to represent traffic networks, and employed CNN to capture the spatial correlations in the traffic data. These approaches then use RNN or additional CNN to model the temporal dependencies in the traffic time series data (Li et al., 2020; Zheng et al., 2020; Ma, Song & Li, 2020). However, CNN’s are not always applicable in traffic road networks of non-Euclidean nature. To solve the problem, many researchers are turning to GCN-based traffic flow models. For example, DCRNN (Li et al., 2017) models spatial dependencies by wandering in both directions on the traffic road topology graph and then uses GRU to capture temporal correlations. ASTGCN (Guo et al., 2019) models the Spatial-temporal dependence of traffic data with spatial attention and temporal attention. STSGCN (Song et al., 2020) captures the local spatio-temporal correlation by combining spatial and temporal blocks through a local spatio-temporal synchronized graph convolution module. STFGNN (Li & Zhu, 2021) learns both local and global spatio-temporal dependencies by processing data-driven spatial and temporal graphs at different moments in parallel. STGODE (Fang et al., 2021) draws on the Dynamic Time Warping (DTW) (Li, Liang & Wang, 2018a) used by STFGNN to generate semantic adjacency matrices for traffic road topology maps to capture deeper spatio-temporal correlations. In recent works, Z-GCNETs (Chen, Segovia & Gel, 2021) introduce the concept of zigzag persistence in the traffic network diagram structure and integrate it into GCNs to enhance the stability of the model. TAMP-S2GCNets (Chen et al., 2021) proposes the Euler Poincare surface for learning the topological structure of traffic network through topological features under the multi-parameter persistence of traffic flow data, and constructs a hypergraph convolution network to model the spatio-temporal dependence of traffic flow. STG-NCDE (Choi et al., 2022) designs two neural control differential equations dealing with temporal and spatial dependencies, respectively, and integrates both to capture spatio-temporal dependencies simultaneously. DSTAGNN (Lan et al., 2022) proposes a spatio-temporal perception distance to dynamically learn Spatial-temporal dependence.

Compared to the spatio-temporal prediction models mentioned above, our proposed STCGAT utilizes a node adaptive learning graph attention network to model the spatial dependency without relying on the topological structure information of the traffic network. Additionally, a bidirectional spatio-temporal component is introduced, which recursively integrates the causal spatio-temporal correlations at different moments within the traffic network. Simultaneously, it captures the global spatio-temporal dependencies present in the overall time series.

Methods

Problem formulation

Traffic flow prediction uses historical traffic flow on the road to predict traffic condition information in the future period.

Definition 1: The graph G=(V,E) is utilized in this study to represent the topological information of the traffic roads. Where V={v1,v2,⋯,vN} is the set of all nodes of graph G, N is the number of all road nodes, and E denotes the set of connected edges of all nodes of graph G.

Definition 2: The historical traffic flow information of a time length T is represented in this study by a feature matrix X={X1,X2,⋯,XN}∈RN×T×F. Where F denotes the feature dimension, Xt={x→t:1,x→t:2,⋯,x→t:N}∈RN×F denotes the set of traffic information of all road nodes at any t time, and x→t:i∈RF then denotes all the feature vectors of node vi.

Traffic flow prediction aims to use the G and X to predict the traffic flow Y′=[Xt+1′,Xt+2′,⋯,Xt+T′] in the following T moments by learning a function f(⋅).

(1) Y′=f(G;(Xt−T,Xt−(T−1),⋯,Xt))

Spatial dependency modeling

In the spatial dimension, the traffic condition data of different road nodes are strongly and dynamically interconnected. However, traditional graph neural networks rely on predefined adjacency matrices to perform graph convolution operations based on factors like connectivity or distance between graph nodes. Although these matrices intuitively represent the positional relationship between nodes, they cannot capture the dynamic spatial correlation between road nodes at different moments. To address this issue, as shown in Fig. 1B, a node adaptive learning mechanism is adopted in this study. This mechanism effectively learns the dynamic correlation information among road nodes at different moments to generate the traffic subgraph Gad specific to the corresponding moment. As shown in the following equation, this mechanism generates the adjacency matrix At~∈RN×N for any moment t.

Figure 1 The STCGAT framework in its entirety.

(2) A~t=softmax(ReLU(EAt⋅EAtT))

where EAt∈RN×d is the embedding dictionary encoding each traffic road node, d is the embedding dimension, EAtT is the transpose matrix of EAt. ReLU is the nonlinear activation function, and softmax is the normalization function.

To capture the adaptive and dynamic spatial dependencies among nodes in the spatial dimension, a NAL-GAT model is proposed in this study. This model integrates a node-adaptive learning mechanism with GAT. Specifically, at any moment t, NAL-GAT computes attention coefficients of neighboring nodes for the node correlation information generated by the node-adaptive learning mechanism. This enables the model to extract the spatial features of traffic roads and aggregate spatial dependencies among the nodes on the graph at moment t. As shown in the equation below, the attention coefficient eijt between node vi and its neighbor node vj is computed in a vertex-wise manner.

(3) eijt=a(Wx→t:i,Wx→t:j)

where a is the computational function of the attention mechanism, and W∈RF×F′ is the graph’s weight matrix of all nodes. The attention coefficient αijt of the graph’s attention layer is then generated by normalizing the attention coefficients of node vi’s neighbors.

(4) αijt=softmaxj(eijt)=exp(eijt)∑k∈A~itexp(eikt)

where A~it denotes all the neighbor nodes of vi.

In addition, it is noted that in the GAT, all nodes share the same parameter space W∈RF×F′. However, this can result in a large graph W when there are more nodes, making the model difficult to optimize. To address this problem, a shared weight pool Wp∈Rd×F×F′ is constructed, which can get the weight matrix W′=EAt⋅Wp∈RF×F′ of each node according to the node’s embedding dictionary EAt.

(5) eijt=LeakReLu(a→T[EAt⋅Wpx→t:i∥EAt⋅Wpx→t:i])eikt=LeakReLu(a→T[EAt⋅Wpx→t:i∥EAt⋅Wpx→t:k])αijt=exp(eijt)∑k∈A~itexp(eikt)

where a→∈R2F′ is the weight matrix, ∥ denotes the connection operation, and LeakReLu is the nonlinear activation function.

To capture deeper feature information, this article further uses the multi-head attention to model spatial dependence. As shown in the following equation, Q sets of mutually independent attention mechanisms are invoked.

(6) x→t:i′=∥q=1QLeakReLu(∑k∈A~itαikt,q(EAtq⋅Wpq)x→t:k)

where αikt,q is the weight coefficient computed by the attention mechanism of the qth group at time t, EAtq⋅Wpq is the weight matrix of the corresponding group, and xi→′∈RQF′ is the new feature representation obtained by passing node vi through the attention layer of the multi-headed graph.

When employing multi-headed attention, it is important to avoid outputting too many features in the network’s last layer. To address this, a separate self-attentive mechanism is utilized to limit the output feature length of each node. Specifically, a new weight pool Wp′∈Rd×QF′×F′′ is employed to map the node’s output dimension from RQF′ to RF′′ and obtain the final output result x→t:i′′∈RF′′.

(7) x→t:i′′=LeakReLu(∑k∈A~itαikt′(EAt′⋅Wp′)x→t:k′)

After completing the graph attention layer operation for all nodes in the graph, the output features can be obtained as Xt′′={x→t:1′′,x→t:2′′,⋯,x→t:N′′}∈RN×F′′. For the convenience of presentation, this process can be expressed in the following equation.

(8) Xt′′=σ(A~tXt(EAt⋅Wp))

where σ(⋅) is the computation function for the graph attention layer.

Local causal spatial-temporal dependency modeling

There is a correlation between traffic conditions in the time dimension at various times. As shown in Fig. 1C, the gating unit of GRU is replaced with NAL-GAT in order to model Spatio-temporal correlations. Specifically, the spatially dependent time series data Xt′′ at any moment t is used as the input data of the GRU.

(9) zt=σ(A~t[Xt,h→t−1](EAtz⋅Wpz))rt=σ(A~t[Xt,h→t−1](EAtr⋅Wpr))ht~=tanh(A~t[Xt,rt⊙h→t−1](EAth~t⋅Wph~t)ht→=zt⊙ht−1+(1−zt)⊙ht~

where h→t−1 is the output at the previous moment, h~t is the candidate hidden layer state, [⋅] is the concat operation in the feature dimension, ht→∈RN×F′′ is the output at the moment t, and ⊙ is the multiplication by elements.

It is important to note that as the input time length rises, so does the network depth of the model. However, the deep network may lead to issues such as gradient disappearance and overfitting in the model. Therefore, the residual module is used to connect the layers of the network in order to enhance the model’s capacity for long-term capture.

(10) zt=σ(A~t[Xt,h→t−1′](EAtz⋅Wpz))rt=σ(A~t[Xt,h→t−1′](EAtr⋅Wpr))ht~=tanh(A~t[Xt,rt⊙h→t−1′](EAth~t⋅Wph~t)ht→=zt⊙ht−1′+(1−zt)⊙ht~h→t′=ε(ω1⊗Xt+ω2⊗ht→)

where ω1 and ω2 are both one-dimensional convolution kernels, ε is the nonlinear activation function, ⊗ denotes the convolution operation, and h→t′∈RN×F′′ is the output of residual concatenation. Until the completion of the above operations at the Tth time step, the sequence data containing the spatio-temporal dependence can be obtained as H→′∈RN×T×F′′.

In addition, traffic data are not always sequentially correlated, and there are complex causal correlations between traffic events. Therefore, bidirectional GRU is utilized to capture the local causal and temporal relationships. The reverse operation is similar to the above operation, and the output results are finally stitched to obtain the output H∈RN×T×2F′′.

(11) H=H→′∥H′←

Global spatial-temporal dependency modeling

From the above procedure, it is evident that GRU is processed by progressively unfolding along the timeline, which causes the output at the present time to depend on the state at the previous time and so lacks the capacity to capture global temporal dependence. A parallel TCN is deployed along the time axis to enhance the performance of extracting long-term Spatio-temporal dependencies. As shown in the following equation, H′∈RN×(T∗2F′′) is utilized as the TCN’s input data. In the time series convolution process, the time series data Hi:′∈R(T∗F′′) of any node vi and a filter f:{0,⋯,l−1}→R are first extended for the elements s.

(12) F(s)=(Hi:′∗df)(s)=∑i=0l−1f(i)⋅Hi:′(s−d⋅i)

where l is the filter size, and s−d⋅i denotes the direction of the timeline past, d is the dilation factor. When d=0, the dilation convolution becomes a regular convolution. In addition, as shown in Fig. 1D, the TCN needs to perform a series of transformations such as Weight Norm and Dropout and use the residual join to obtain the output o∈RN×(T∗2F′′) as in the equation below.

(13) o=Activation(x+F(x))

The prediction layer

Afterward, a two-layer fully connected neural network is utilized to perform a linear transformation specifically tailored to the dimensions of the output sequence.

(14) Y′=W2⋅φ(W1⋅o+b1)+b2

where b1 and b2 are the bias values, W1∈RF′′′×(T∗2F′′) and W2∈R(T×F)×F′′′ are the weight matrices, and Y′∈RN×T×F is the prediction result.

During model training, the optimization is performed using the L1 loss function and the Adam optimizer in order to minimize the error between the predicted Y′ and the labeled values Y=[Xt+1,Xt+2,⋯,Xt+T]. In summary, the specific calculation process of STCGAT is shown in Algorithm ??.

(15) loss=1T∑i=1T|Yt+i−Yt+i′|

Datasets

Extensive experiments were carried out on four publicly available transportation datasets namely PeMS03, PeMS04, PeMS07, and PeMS08 (Guo et al., 2021). These datasets were obtained from Caltrans’ Performance Measurement System (PeMS), which has deployed over 39,000 traffic detectors on California freeways to collect real-time traffic data and aggregates the collected data every 5 min. Details about the datasets are provided below. PeMS03: The dataset contains 358 sensors connected by 547 edges, recording 26,208-time steps of traffic data.

PeMS04: The dataset contains 307 sensors connected by 340 edges, recording 16,992-time steps of traffic data.

PeMS07: The dataset contains 883 sensors connected by 866 edges, recording 28,224-time steps of traffic data.

PeMS08: The dataset contains 170 sensors connected by 295 edges, recording 17,856-time steps of traffic data.

Baseline methods

To evaluate the prediction performance of the proposed model, a series of baseline models are selected for comparative experiments in this study. These baseline models encompass the mainstream traffic flow prediction models outlined in this article, namely HA (Duan et al., 2016), LSTM (Karimzadeh et al., 2021), DCRNN (Li et al., 2017), ASTGCN (Guo et al., 2019), STSGCN (Song et al., 2020), STFGNN (Li & Zhu, 2021), STGODE (Fang et al., 2021), Z-GCNETs (Chen, Segovia & Gel, 2021), TAMP-S2GCNets (Chen et al., 2021), STG-NCDE (Choi et al., 2022) and DSTAGNN (Lan et al., 2022). Furthermore, a traffic flow prediction network called AGCRN (Bai et al., 2020), which incorporates an adaptive graph convolution module, is also chosen as a new baseline model.

Results and discussion

Experimental settings

The used dataset is divided into 60% training set, 20% validation set, and the remaining 20% test set with Z-score standardization. The partitioned dataset is then processed through a sliding window of length 2T, where the first T time lengths of serial data are used as historical data, and the last T time lengths of data are used as labeled values. In our experiments, T was set to 12.

The STCGAT model was implemented using the PyTorch deep learning framework, with hyperparameters set as follows: the node embedding feature dimension was set to 10, the hidden layer size was 64, the model used three multi-head attention mechanisms and a convolutional kernel size of two. During the training process, a batch size of 64 was utilized, along with an Adam optimizer and a learning rate of 0.001. The model was trained for a maximum of 300 epochs. The experiments were carried out using a Linux server equipped with an NVIDIA GeForce 2080Ti GPU.

In order to evaluate the predictive accuracy of both the proposed model and other baseline models, the following performance metrics were employed as evaluation criteria. Mean absolute error (MAE): MAE=1L∑i=1L⁡|Yi−Yi′|

Root mean squared error (RMSE): RMSE=1L∑i=1L(Yi−Yi′)2

Mean absolute percentage error (MAPE): MAPE=100%L∑i=1L⁡|Yi−Yi′Yi|

where L is the total number of samples, all models were subjected to five experiments, and the mean was taken as the final experimental result.

Experiment analysis

On PeMS03, PeMS04, PeMS07, and PeMS08, our model was compared with the twelve representative baseline approaches described previously. Table 1 shows the prediction performance results of STCGAT and other baseline models within 1 h (12 prediction steps). It can be observed that: (1) In traffic flow prediction tasks, the effectiveness of GCN in explicitly modeling spatial correlation is demonstrated by the superior performance of the GCN-based approach compared to the LSTM-based time series forecasting methods. This highlights the significance of incorporating spatial correlation into the prediction models; (2) The performance metrics of our improved GAT-based method on each dataset are almost better than other advanced baseline models, achieving significant results; (3) as shown in Table 2, STCGAT is compared with several advanced spatio-temporal prediction models for the purpose of parameter count and training cost comparison. The results show that STCGAT achieves the best prediction performance with relatively reasonable control of the overall parameter number and training cost compared with the current advanced spatio-temporal prediction models; (4) as shown in Fig. 2, the prediction results are visualized by continuously saving 24 prediction snapshots of STCGAT and other advanced baseline models on the test set. It can be observed that STCGAT demonstrates faster and more accurate recovery of prediction accuracy, particularly in the presence of missing data. The corresponding changes during peak traffic periods are also more accurate; (5) as shown in Fig. 3, which demonstrates the comparison of the prediction power of STCGAT with other spatio-temporal prediction models on different horizon, it can be observed that the performance curve of STCGAT exhibits relatively small oscillations on each dataset. This indicates that the proposed method possesses stable short- and long-term spatio-temporal prediction capabilities.

Table 1 Comparative experimental results of STCGAT and other baseline models.

Model	Dataset	PeMS03	PeMS04	PeMS07	PeMS08	
	Metrics	MAE	MAPE	RMSE	MAE	MAPE	RMSE	MAE	MAPE	RMSE	MAE	MAPE	RMSE	
HA	32.04	32.96%	51.42	39.57	27.87%	58.84	44.92	24.02%	65.47	35.47	27.95%	59.14	
LSTM	20.96	20.76%	36.01	25.01	16.18%	41.42	33.26	14.32%	59.92	23.49	14.55%	38.89	
ASTGCN	17.65	16.94%	29.63	22.03	14.59%	34.99	24.01	10.73%	37.87	18.36	11.25%	28.31	
DCRNN	17.48	16.83%	29.19	21.22	14.17%	33.44	24.69	10.80%	37.88	16.82	10.92%	26.32	
STSGCN	17.48	16.78%	29.21	21.19	13.90%	33.65	24.26	10.21%	39.03	17.13	10.96%	26.80	
STFGNN	16.77	16.30%	28.34	20.18	13.94%	32.41	22.07	9.21%	35.80	16.64	10.60%	26.25	
AGCRN	15.97	15.23%	28.11	19.83	12.97%	32.26	21.13	8.96%	35.20	15.95	10.09%	25.22	
STGODE	16.43	16.43%	27.43	20.84	14.82%	32.56	22.87	10.24%	37.64	16.72	10.57%	26.01	
Z-GCNETs	16.44	16.39%	28.52	19.51	13.02%	31.90	21.77	9.22%	34.90	16.13	10.11%	25.90	
TAMP-S2GCNets	16.19	15.32%	27.72	20.53	13.03%	34.44	22.23	9.82%	35.76	16.90	10.45%	27.30	
STG-NCDE	15.58	15.32%	27.08	19.81	14.04%	31.57	21.78	9.42%	34.75	15.62	10.14%	24.87	
DSTAGNN	15.54	14.69%	27.13	19.42	12.78%	31.75	21.91	9.31%	35.57	15.96	10.04%	25.21	
STCGAT	15.31	14.90%	27.03	19.21	12.36%	31.12	20.89	8.79%	34.00	15.41	9.78%	24.61	
Notes:

1. The result highlighted by the underline represents the best outcome obtained among all the baseline models in the experiment.

2. The experimental outcomes of our proposed models are indicated in bold.

Table 2 Models parameters and training cost statistics.

Model	PeMS04	PeMS07	PeMS08	
	Parameters	Train time (epoch)	Parameters	Train time (epoch)	Parameters	Train time (epoch)	
AGCRN	748,810	59.08 s	754,570	324.43 s	150,112	33.89 s	
Z-GCNETs	455,034	77.68 s	460,794	664.32 s	453,664	63.34 s	
TAMP-S2GCNets	1,189,992	525.82 s	1,201,512	6,434.23 s	1,187,252	328.64 s	
STG-NCDE	2,550,024	658.42 s	2,561,544	3,149.18 s	2,547,284	379.89 s	
DSTAGNN	3,579,728	273.34 s	14,353,744	2,448.03 s	2,296,860	133.97 s	
STCGAT	1,410,730	82.73 s	1,416,490	492.68 s	1,409,360	42.11 s	

Figure 2 Diagram of model prediction results when data is missing and data peaks on PeMS03, PeMS04, PeMS07 and PeMS08 datasets.

(A) and (C) are model predictions when data are missing. (B) and (D) are model predictions when data are at their peak.

Figure 3 The MAE and MAPE on each horizon.

Ablation study

Four STCGAT-based model variants were created in this study, and STCGAT was contrasted with these variants to understand the influence of STCGAT’s numerous modules on its predictive performance. The differences between these four variants of the model are described below. 1. w/o reverse GRU: The model models the time dependence using only positive GRU.

2. w/o ResNet: The model eliminates the STCGAT residual connection module.

3. w/o node embedding: The model uses a predefined adjacency matrix to replace the self-generated adjacency matrix in STCGAT.

4. w/o TCN: The model removes the TCN from STCGAT.

Table 3 presents the experimental results conducted with the aforementioned variants and STCGAT. Combined with the histogram of evaluation metrics for each variant model at different time steps in Fig. 4, the following observations can be made: (1) Each evaluation metric of w/o TCN is at the maximum value and the performance metrics are similar for short term (15 mins) and long term (60 mins). It illustrates the necessity of using temporal convolutional networks to extract the global temporal correlation of traffic flow; (2) the performance metrics of w/o node embedding decrease significantly when the node self-learning module is removed. This proves that learning the dynamic spatial correlation among nodes from the state information of traffic flow at different moments better expresses how the traffic flow dynamics change in reality; (3) the performance metrics values of w/o ResNet compared to STCGAT are significantly larger, i.e., removing the residual module has a more significant impact on both datasets. This indicates that the application of the residual module helps STCGAT to mitigate the problem of overfitting or gradient disappearance caused by the superposition of network layers to a certain extent; (4) the performance metrics of with or without reverse GRU also increase on both the PeMS04 and PeMS08 datasets, indicating that effectively capturing the causality of traffic flow data helps to analyze the spatio-temporal correlation of traffic flow more comprehensively; (5) compared with the four variants, STCGAT has the best performance. This statement highlights two key observations regarding the performance of STCGAT. Firstly, it underscores the significance of individual modules within the STCGAT framework. Secondly, it demonstrates the superior ability of STCGAT to effectively capture and extract spatio-temporal correlations present within traffic flow series.

Table 3 Results of ablation experiments of STCGAT.

Model	Dataset	PeMS04	PeMS08	
	Metrics	MAE	MAPE	RMSE	MAE	MAPE	RMSE	
w/o reverse GRU	19.80	13.29%	31.64	15.98	10.33%	25.05	
w/o ResNet	19.72	13.57%	32.45	16.01	10.31%	25.37	
w/o node embedding	22.03	15.29%	34.65	17.68	11.04%	27.65	
w/o TCN	23.74	16.31%	39.05	21.91	13.84%	34.76	
STCGAT	19.21	12.36%	31.12	15.41	9.78%	24.61	

Figure 4 Short-term and long-term prediction performance on the PeMS04 and PeMS08 datasets.

Conclusion

This article proposes the spatio-temporal causal graph attention network (STCGAT). Firstly, STCGAT encodes road nodes into node embeddings and adaptively learns the relationships between nodes based on the traffic conditions at different moments, without relying on predefined adjacency matrices. This is integrated into the GAT to form the Node-Adaptive Learning GAT (NAL-GAT) to model the spatial dependencies dynamically. Secondly, STCGAT reconstructs NAL-GAT into a GRU to capture local spatio-temporal dependencies. The bi-directional GRU is used to capture spatio-temporal causality at a fine-grained level. Additionally, STCGAT introduces a residual module to reduce network degradation caused by deep networks. Finally, STCGAT utilizes the temporal convolutional network to capture global spatio-temporal dependence information by processing time series data in parallel. Extensive experiments on multiple datasets demonstrate that STCGAT outperforms advanced spatio-temporal prediction models, with excellent spatio-temporal modeling capability for highly nonlinear traffic data. Future work will investigate the proposed model in other spatiotemporal data mining problems, such as weather spatiotemporal data mining tasks.

Supplemental Information

Supplemental Information 1 Source Code.

Click here for additional data file.

Supplemental Information 2 PeMS03 Dataset.

Click here for additional data file.

Supplemental Information 3 PeMS08 Dataset.

Click here for additional data file.

Additional Information and Declarations

Competing Interests

Author Contributions

Data Availability

The authors declare that they have no competing interests.

Wei Zhao conceived and designed the experiments, authored or reviewed drafts of the article, and approved the final draft.

Shiqi Zhang conceived and designed the experiments, performed the experiments, analyzed the data, performed the computation work, prepared figures and/or tables, authored or reviewed drafts of the article, and approved the final draft.

Bei Wang performed the experiments, performed the computation work, prepared figures and/or tables, and approved the final draft.

Bing Zhou performed the experiments, analyzed the data, authored or reviewed drafts of the article, and approved the final draft.

The following information was supplied regarding data availability:

The code is available at Zenodo: zsqZZU. (2023). zsqZZU/STCGAT: v1.0.0 (v1.0.0). Zenodo. https://doi.org/10.5281/zenodo.7811804.

The data is available at Zenodo: Shiqi Zhang. (2023). TrafficDataSets (V1.0) [Data set]. Zenodo. https://doi.org/10.5281/zenodo.7816008.

The baseline models are available at Zenodo: zsqZZU. (2023). zsqZZU/STCGAT_Baseline: STCGAT_Baseline (STCGAT_Baseline). Zenodo. https://doi.org/10.5281/zenodo.7833114.

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
