# Peer review of "Spatio-temporal causal graph attention network for traffic flow prediction in intelligent transportation systems"

_PeerJ Computer Science, doi:10.7717/peerj-cs.1484_

## Round 0.1 · original submission · Minor Revisions

Dear authors,

Your paper has been reviewed by two reviewers who asked for revisions of the paper. Please revise the paper according to comments by reviewers, mark all changes in a new version of the paper, and provide a cover letter with replies to them point to point.

Reviewer 1 ·

Basic reporting

It is a very interesting research, based on an intelligent transportation systems...

Experimental design

The paper represents an extremely important segment of research in road engineering. The abstract of the paper describes the problem extremely well and indicates the objectives. The authors present an excellent literature review.

Validity of the findings

Special praise to the author for the methodology of the paper and the given case study. To make the work even better, I give certain suggestions:

In the title of the paper, remove the abbreviation STCGAT. Abbreviations are not placed in the title.
Line 17 in the abstract- It is necessary to specify which methods... or modify the sentence.
Chapter introduction, it is necessary to divide it into 2 chapters (Introduction and literary review)
Figure 1. - I am of the opinion that the figure is redundant in the representation of the paper.
Figure 2 - The authors explain the model in the image in the name of the figure. State the name of FIGURE 2, and explain the rest in the text
Line 150 - Add 1-2 more sentences here with an explanation about ... Spatio-temporal component to capture both local and global Spatio-temporal causality.
Line 203-228- I suggest the authors make a table
In some parts of the paper, the first person plural is used to write the paper (82th,84th,86th,89th, 91th, 93th 94th 149th 155th 158th 165th 162th 166th 171th 173th 176th 179th 180th 181th 182th 184th 201th 230th 250th 255th 261th 263th 264th 268th 276th 277th). That needs to be corrected. The paper can not be written in the first person!

Additional comments

No comment

·

Basic reporting

The title and abstract are in compliance with the research presented in the paper. Paper is written from scientific point.
I suggest that the structure of the paper be standardized. The title of the second chapter should be "LITERATURE REVIEW" instead of "RELATED WORKS". In accordance with the content, I propose to rename the fourth chapter "EXPERIMENT" to "RESULTS and DISCUSSION".

Also, the authors should consider moving the subchapters "Datasets" and "Baseline Methods" to the third chapter "METHODS".

In the subchapter "Baseline Methods" the authors repeat the information that was presented in the previous chapters. I suggest that the baseline models should be presented in detail in the "LITERATURE REVIEW" chapter, while in subchapter "Baseline Methods" the authors should only list the references without further description.

The title of figure 2 is too long. I suggest a shorter title, for example "Figure 2. The STCGAT framework in its entirety", and the remaining part of the text should be formatted as one paragraph within the paper.

Reference "Cascetta, 2013" (line 45) - not listed in REFERENCES chapter. Other references are relevant and up to date.

Experimental design

This article addresses interesting subject within Aims and Scope of the journal. The authors proposed a solution for predicting traffic flow on roads and named it Spatio-temporal prediction framework based on Graph Attention Network. Their solution offers a data-driven graph structure learning approach that autonomously learns the relationship information between road nodes and models the spatial correlation of traffic networks without relying on the traffic network graph structure information. The models are presented in detail so that the research can be replicated.

Validity of the findings

The authors stated the conclusions based on the obtained results and connected them with the original research question.

---

## Round 0.2 · accepted · Accept

Dear authors,

Both reviewers have accepted a revised version of the paper.

Reviewer 1 ·

Basic reporting

No comment

Experimental design

No comment

Validity of the findings

No comment

Additional comments

No comment

·

Basic reporting

The authors have corrected the paper in accordance with the comments.

Experimental design

no comment

Validity of the findings

no comment